# The Effects of Temperature on the Turnover of $\delta^{18}$O and $\delta$D in Juvenile Corn Snakes (*Elaphe guttata*): A Novel Study with Ecological Implications

**Samuel J. Hirt [1,*] and Kent A. Hatch [2]**

[1] Division of Natural Sciences, Southern Virginia University, 1 University Hill Drive, Buena Vista, VA 24416, USA

[2] Department of Biological and Ecological Sciences, Long Island University Post, 720 Northern Blvd., Brookville, NY 11548, USA; kent.hatch@liu.edu

[*] Correspondence: samuel.hirt@svu.edu

**Abstract:** The use of natural variation in stable isotope ratios continues to be used in ecological studies without proper validation through laboratory studies. This study tested the effects of temperature, time, and turnover in the scales of juvenile corn snakes (*Elaphe guttata*) in a controlled, laboratory environment. Snakes were assigned to four treatment groups (24 °C, 27 °C, 30 °C, and freely thermoregulating), and one snake from each group was sacrificed weekly. Scales from each snake were washed, dried, and analyzed for $\delta$D and $\delta^{18}$O at the Stable Isotope Research Facility for Environmental Research at the University of Utah. The effects of temperature on the turnover of tissues was only significant when comparing the thermoregulating group to the pooled treatment groups (24 °C, 27 °C, and 30 °C) in the $\delta^{18}$O of scales ($p$ = 0.006). After normalizing data on the $\delta$D and $\delta^{18}$O using percent change for comparison, $\delta^{18}$O appeared to be turning over at a faster rate than $\delta$D as indicated by an analysis of covariance (ANCOVA) test for homogeneity of slopes ($F_{1,53}$ = 69.7, $p$ < 0.001). With further testing of assumptions, a modification of our methods could provide information on the composition of drinking water sources in a species that switches between two isotopically distinct sources, such as during seasonal shifts in habitat or migration, and/or estimates of long-term field metabolic rates based on the turnover of these isotopes.

**Keywords:** stable isotopes; ectotherm; turnover; validation; non-invasive; deuterium; oxygen

## 1. Introduction

Stable isotopes are useful as natural, biological indicators, providing ecological data such as patterns of migration, nutrition, and trophic position [1,2]. The observance of naturally occurring variations in stable isotopes continually provides application in clinical [3], forensic [4,5], and ecological fields of study, especially among mammals and birds [6–9]. Applying stable isotope analysis to reptiles is less common; however, applications are becoming more frequent [10,11].

The significance and the interpretation of isotopic data need validation by controlled, laboratory experiments before application in ecological studies. Despite arguments that validation research should be done before application in the field, there continues to be a lack of studies providing these important background data [10,12–14]. Without validation studies, results from field studies will continue to be based on untested assumptions, and contribute to inaccurate interpretations. In this study, we provide results from controlled laboratory experiments that can provide information for the interpretation of future field experiments, especially with terrestrial ectothermic vertebrates.

Ectotherms are limited in the behavioral regulation of their body temperature and metabolism by the environmentally available temperatures. Ambient temperature affects the rate at which ectotherms

are able to incorporate ingested food and water. Therefore, ectotherms behaviorally modify their body temperature by inhabiting microhabitats that maximize their performance [15,16], reducing or eliminating the metabolic demand to maintain a constant and continually elevated body temperatures. This allows them to generally have a more efficient metabolism than endotherms, and therefore generally have lower turnover rates in tissues than endothermic organisms [17].

Understanding the rates of turnover of stable isotopes in various tissues is a necessary component for interpreting ecological data from studies in the field. Experimentally, the temperature at which ectotherms and endotherms are held affects the turnover rates of stable isotopes in tissues differently [18–20]. Lowering the temperature in ectotherms below their thermoneutral threshold will cause a decrease in metabolic rate, whereas lowering the temperature of endotherms causes them to elevate their metabolic rate to maintain their internal body temperature. In experiments where endotherms were held at a constant temperature, effects on stable isotope turnovers varied depending on species, isotope, and tissue [20,21]. The effect of a raised or decreased temperature, however, is yet to be experimentally determined in the turnover of oxygen and hydrogen isotopes in any species of ectotherm. The effect of temperature is ecologically significant when interpreting field-collected samples of $\delta D$ and $\delta^{18}O$, especially when sampling among different and changing conditions such as are found in temperate biomes. Using controlled temperature chambers, we expected to find a significant difference in the isotopic turnover of tissues of ectotherms held at different, but constant temperatures, because of their reliance on ambient temperature to control metabolic rate.

While endogenous carbon and nitrogen come entirely from ingested food, the hydrogen and oxygen in tissue of organisms come from multiple sources. The difference in the sources of hydrogen isotopes (food and water) and oxygen isotopes (food, water, and air) causes differences in the rates of turnover [22,23]. The difference in the rate of turnover in $\delta D$ and $\delta^{18}O$ isotopes is used to determine field metabolic rates using the doubly labeled water technique [24]. Similarly, metabolic activity could be estimated by analyzing $\delta D$ and $\delta^{18}O$ in tissues before and after an organism naturally switches between isotopically distinct water sources, but the laboratory assumptions of this method remain to be tested. The aims of this study include testing assumptions particular to temperature and the turnover rate of $\delta D$ and $\delta^{18}O$.

The effects of temperature on metabolism should be detectable in the turnover rate of $\delta D$ and $\delta^{18}O$ in scales of juvenile corn snakes (*Elaphe guttata*). Turnover in nitrogen and carbon isotope ratios of captive juvenile corn snakes showed variable results based on carbon or nitrogen turnover, the enrichment or depletion of isotopes, and the type of tissue [25]. We tested the hypothesis that $\delta D$ and $\delta^{18}O$ in scales of snakes will turn over at a faster rate when snakes are held at higher temperatures.

## 2. Materials and Methods

We used corn snakes as our model ectotherm because they are common, readily available, and are easily kept in captivity. Corn snakes were obtained from the University of Texas at Tyler in March 2005. We immediately divided the snakes into three temperature treatments: their preferred body temperature 27 °C [26], 24 °C, and 30 °C. We also had a separate control group in which snakes were allowed to freely thermoregulate. Snakes of the same clutch were distributed among different treatment groups to control for genetic bias.

All individuals were housed separately and the location of their housing within the environmental chambers was changed at random to eliminate differences due to temperature or light gradients within cabinets. Each snake in a temperature-controlled group was housed in a plastic container measuring 8 × 12 × 4 inches, with holes in the lids for ventilation and wood chips within the container to provide some structural habitat and refugia. The same type of lighting (fluorescent bulbs) was used within the temperature cabinets and in the freely thermoregulating group. Lights were put on a 12-hour light and dark alternating timer. The lights in the temperature cabinets were positioned along the sides of the cabinet extending vertically, while the lights of the thermoregulating group were placed on top of the cages extending horizontally. Each snake in the thermoregulating group was kept in a glass aquarium

measuring $8 \times 12 \times 24$ inches, with a heating pad on half of the bottom of the cage. The available temperatures within each thermoregulating cage ranged from 33.9 °C at the surface (36.6 °C underneath the sand) nearest the heat source to 24.9 °C at the surface (24.6 °C underneath the sand) farthest from the heat source. The snakes were able to move freely between the substrate of these temperatures.

Since all the snakes were originally from Tyler, Texas, they were raised on tap water local to Tyler ($\delta D = -0.5‰$ (+/− 1.0), $\delta^{18}O = -0.85‰$ (+/− 0.40). Once moved to Brigham Young University, Provo, Utah, we took pains to ensure that both the dietary and drinking water signals had their source in local, Provo, Utah tap water, ($\delta D = -119.8‰$ (+/− 1.0), $\delta^{18}O = -16.15‰$ (+/− 0.40)). The access to drinking water of snakes was limited to a half an hour daily by filling a petri dish with Utah tap water and placing within their housing structure. Limiting the exposure of water to the environment would reduce any fluctuation in the isotopic value of the water due to evaporation. To control intake of food, all snakes were fed one pinkie per week. Both adult mice and their pups were raised on local Provo, Utah tap water.

At week 0, one snake from each temperature group was sacrificed as a control. We sacrificed one snake per group per week for eight weeks and stored specimens in a −80 °C freezer. Mass at death was also measured before euthanasia, but we neglected to make mass measurements throughout the study; thus, an analysis on the change in mass was not possible. Snakes were randomly assigned to be sacrificed before the study began. However, not all 40 snakes were included in the final analyses. Snakes that died or escaped their housing and were recaptured were not included in the analyses. Because of the deaths and escapes, we had $n = 6$ for the thermoregulating group, $n = 8$ for the 24 °C group, $n = 10$ for the 27 °C group, and $n = 9$ for the 30 °C group. Procedures for housing and sacrifice were done under the approval of the Brigham Young University Institute for Animal Care and Use Committee IACUC (pn # 041205).

Scales were collected from thawed specimens as follows: a one-inch section of the tail was cut and removed from the rest of the body. From the tail section, scales were pulled away from muscles and connective tissue, then put in marked centrifuge tubes. The scale samples were then rinsed twice, firstly with petroleum ether to remove all lipids and other impurities, and secondly with distilled water. After rinsing, the samples were put in a 40 °C dryer for 48 h and then stored in a −80 °C freezer. Scales were chosen to be analyzed because results could be applied to shed skins found in the field and to non-lethal sampling of scales from live snakes.

A 1-mg sample was taken from each tissue or substrate and analyzed for $\delta D$, and $\delta^{18}O$ using isotope ratio mass spectrometry (IRMS). Samples were sent to the Stable Isotope Research Facility for Environmental Research at the University of Utah for analysis. All hydrogen and oxygen isotope ratios are expressed in delta notation and are given relative to Vienna Standard Mean Ocean Water (VSMOW).

$\Delta D$ and $\Delta^{18}O$ were calculated using the control snakes sacrificed at week 0. Four snakes were not given local Provo, Utah water and were sacrificed immediately upon arrival. Values of $\delta D$ and $\delta^{18}O$ calculated from the scales of these snakes were averaged and subtracted from samples taken each week by the snakes assigned to the three treatments and thermoregulating groups to determine $\Delta D$ and $\Delta^{18}O$. $\Delta D$ and $\Delta^{18}O$ were used as our dependent variables.

Effects of temperature and time on changes in $\delta D$ and $\delta^{18}O$ were tested using a multiple linear regression statistical test. Changes in $\delta D$ and $\delta^{18}O$ were calculated by averaging the $\delta D$ and $\delta^{18}O$ values of the snakes sacrificed at week 0 and then subtracting the values from the $\delta D$ and $\delta^{18}O$ values of individual snakes sacrificed each week. We calculated Pearson's correlation tests for the variables temperature, weight at death, and $\delta D$ and $\delta^{18}O$. All statistical analyses were calculated using the Statistical Package for Social Sciences (SPSS) version 23.

We tested for significance of all variables in our linear regression model and excluded models that were not significant. There was no significant interaction between time and temperature between the three treatment groups ($\delta D \ p > 0.30$; $\delta^{18}O \ p > 0.10$); thus, the interaction term was not used in our model. Clutch was also not a significant variable in either linear regression model ($\delta D \ p > 0.70$; $\delta^{18}O$ $p > 0.40$), and was not used in our analyses.

To test if there was a difference in the regression of $\delta D$ and $\delta^{18}O$, data were normalized by taking each data point and subtracting the value of the snakes at week 0. Ranges of $\delta D$ and $\delta^{18}O$ were calculated by taking the difference between $\delta D$ and $\delta^{18}O$ of water in Texas from water in Utah. The $\Delta D$ was divided by the range of $\delta D$ to express data points as a percentage; a similar process was done for $\delta^{18}O$. A univariate analysis of covariance (ANCOVA) with normalized data (percentage) as the dependent variable, isotope as a fixed factor, and week as a covariate was used to determine significant differences in slope, intercept, and means of normalized data.

## 3. Results

Before the experiment was conducted, we measured the $\delta D$ and $\delta^{18}O$ of the drinking water of both locations. The value of $\delta D$ was $-0.5‰$ $(+/- 1.0)$ and that of $\delta^{18}O$ was $-0.85‰$ $(+/- 0.40)$ for the Texas drinking water. The value of $\delta D$ was $-119.8‰$ $(+/- 1.0)$ and that of $\delta^{18}O$ was $-16.15‰$ $(+/- 0.40)$ for the Utah drinking water.

To determine differences in our three temperature groups, we used a multiple linear regression with changes in $\delta D$ and $\delta^{18}O$ as the dependent variables, and temperature and week sacrificed as independent variables. We left the thermoregulating group out of this analysis because snakes were not held at constant temperatures. The model was significant for $\delta D$ ($F_{2,22} = 13.148$, $p < 0.001$; $r^2 = 0.568$). The coefficient for week was significant ($p < 0.001$); however, the coefficient for temperature was not significant ($p = 0.637$). The model was significant for $\delta^{18}O$ ($F_{2,22} = 21.681$, $p < 0.001$, $r^2 = 0.684$). The coefficient for week was also significant ($p < 0.001$); however, the coefficient for temperature was also not significant ($p = 0.679$).

The same linear multiple regression analysis was conducted with the thermoregulating group, and the three different temperature treatments were pooled together as a second group since temperature was not significant in the previous analysis. Using treatment and time as independent variables, and $\Delta D$ as a dependent variable, the linear regression model was significant ($F_{2,27} = 17.315$, $p < 0.001$, $r^2 = 0.581$). Within the model, effect of treatment was not significant ($p = 0.873$) and effect of time (week sacrificed) was significant ($p < 0.001$). The corresponding model using $\Delta^{18}O$ as a dependent variable was significant ($F_{2,27} = 32.281$, $p < 0.001$, $r^2 = 0.721$; see Figure 1). Within the oxygen model, both week sacrificed ($p < 0.001$) and treatment ($p = 0.006$) were statistically significant.

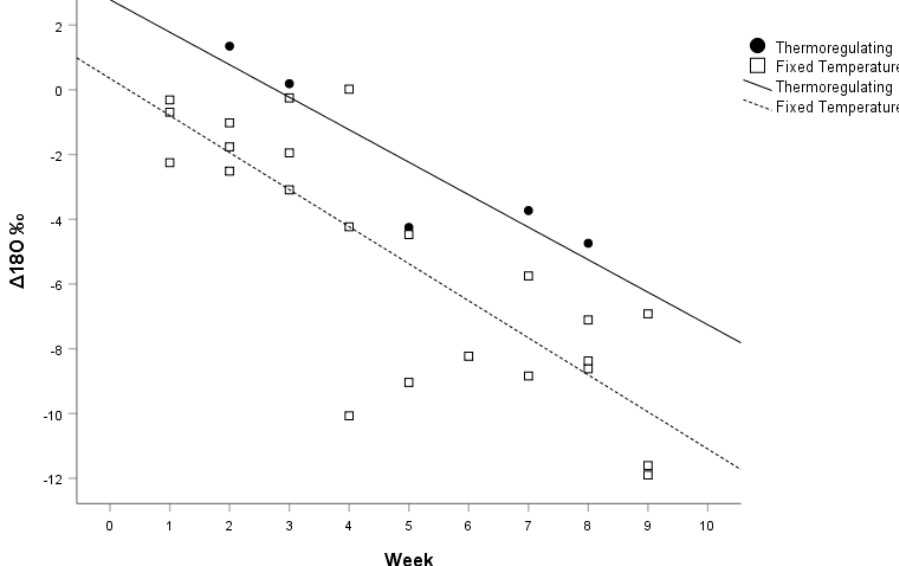

**Figure 1.** The effect of time and treatment on $\Delta^{18}O$ (‰) of scales of corn snakes. $\Delta^{18}O$ was calculated using the average $\delta^{18}O$ from the scales of four snakes sacrificed before the treatments were assigned (thus, these values were representative of the $\delta^{18}O$ of scale tissue for snakes raised on Texas water) and then subtracted from $\delta^{18}O$ of scales from snakes sacrificed each week from the treatment groups.

Because the treatment groups (24 °C, 27 °C, and 30 °C) were not significantly different, they were lumped together into one group (fixed temperature) and are represented by the open squares. The snakes that were allowed to freely thermoregulate (thermoregulating) make up the control group and are represented by closed circles. The effect of time on $\Delta^{18}O$ was significant ($p < 0.001$) and the effect of treatment was also significant ($p = 0.006$). The linear equation for the $\delta^{18}O$ thermoregulating group $y = -1.004x + 2.785$ is represented by the solid line, while the linear regression equation for the constant-temperature groups $y = -1.145x + 0.352$ is represented by the dashed line.

The least-squares regression line of $\delta D$ was $y = 0.00872 - 0.03x$ and the least-squares regression line for $\delta^{18}O$ was $y = 0.05 - 0.07x$ (see Figure 2). Results of the ANCOVA test for homogeneity of slopes found a significant difference in the interaction between isotope and week ($F_{1,53} = 69.7$, $p < 0.001$) indicating a significant difference in the slopes of the two regression lines. The intercept was not significantly different ($F_{1,53} = 0.767$ $p = 0.385$). The means of the isotopes were significantly different ($F_{1,53} = 69.7$, $p < 0.001$).

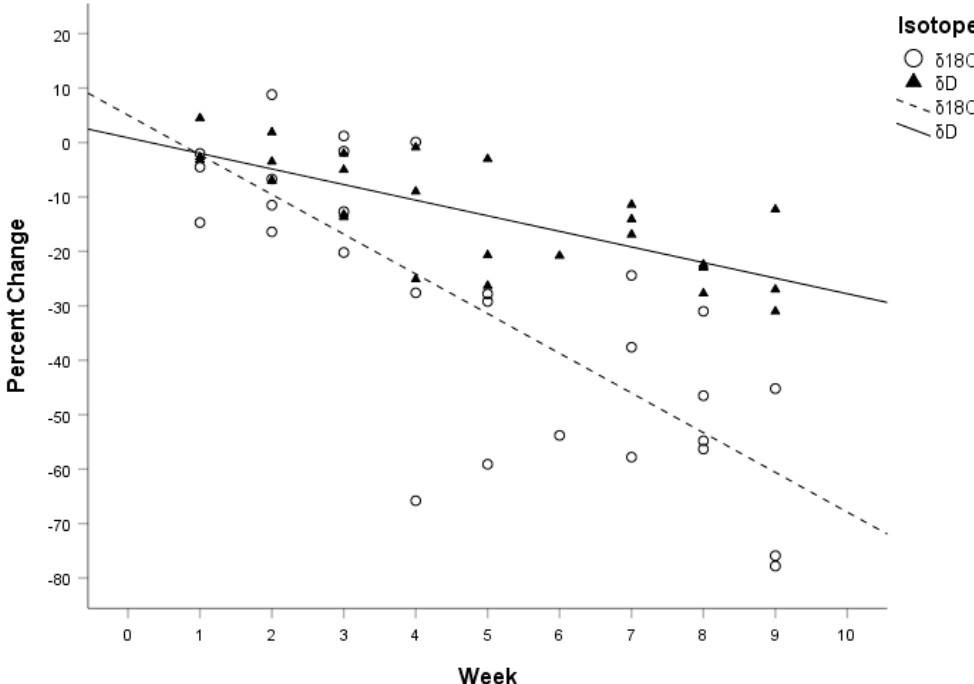

**Figure 2.** The least-squares regression lines for $\delta D$ are indicated by the solid line ($y = 0.00872 - 0.03x$) and triangles and those for $\delta^{18}O$ are indicated by the open circles and dashed line ($y = 0.05 - 0.07x$). Results of the analysis of covariance (ANCOVA) test for homogeneity of slopes found a significant difference in the interaction between isotope and week ($F_{1,53} = 69.7$, $p < 0.001$), indicating a significant difference in the slopes of the two regression lines. The intercept was not significantly different ($F_{1,53} = 0.767$, $p = 0.385$). The means of the isotopes were significantly different ($F_{1,53} = 69.7$, $p < 0.001$). This is consistent with experiments that used doubly labeled water, which showed oxygen isotopes to turn over at a greater rate than hydrogen isotopes.

## 4. Discussion

Oxygen and hydrogen isotopes are underutilized in ecological studies of ectotherms, partly because of the paucity of controlled laboratory experiments needed to effectively interpret data gathered the field [12,13]. Most studies utilizing natural variation in oxygen and hydrogen isotopes on vertebrates were on birds and mammals [6] and not reptiles and amphibians. Our purpose in this laboratory experiment was to provide baseline data using oxygen and hydrogen isotopes that are necessary for application in field studies of ectotherms, particularly snakes. Our results may prove

valuable for field studies interested in taking non-invasive samples such as skin sheds, minimally invasive scale samples from live-caught snakes, or museum specimens.

This experiment was the first to study the effects of temperature and metabolic rate on the turnover of hydrogen and oxygen isotopes in snake tissues. Many studies examined the effects of metabolic rate, diet, size, and growth on nitrogen and carbon isotope turnover in tissue of ectothermic organisms [18,25,27]. However, effects of metabolic rate, diet, size, and growth of $\delta$D and $\delta^{18}$O isotopes still require further investigation as our study indicates. Variables for change in growth were not included in our analyses, which would be an important factor to include in future studies.

As expected, time had a significant effect on the turnover of $\delta$D and $\delta^{18}$O in the scales of our juvenile corn snakes. We expected a turnover of $\delta$D and $\delta^{18}$O in their tissues, because the difference in Texas and Utah water was 119.3‰ for $\delta$D and 15.3‰ for $\delta^{18}$O. Because these were juvenile snakes, we also expected that a greater proportion of their metabolism would be devoted to tissue growth and, thus, tissues would show incorporation of the novel isotope signature in their water and food over time. This growth would be apparent in their scales, which are replaced via shedding during growth. We hoped to see a complete turnover to the novel signature in isotopes within the time period of our study; however, because this kind of study had not been done before, we did not know how long it would take.

The difference between $\Delta$D and $\Delta^{18}$O of drinking water and scales was quite large (See Figures 3 and 4), but consistent with other studies that determined $\Delta\delta$D and $\Delta\delta^{18}$O in keratinous tissues such as hair, nails, claws, or fur. Baboons had $\Delta\delta^{18}$O values in hair that were on average 18–19.8‰ higher than the water from their drinking source [28], compared to a 15.3‰ difference in our study. The $\Delta\delta$D values in Japanese quail feathers were 75‰ lower than their drinking water [21] compared to 119.3‰ in our study.

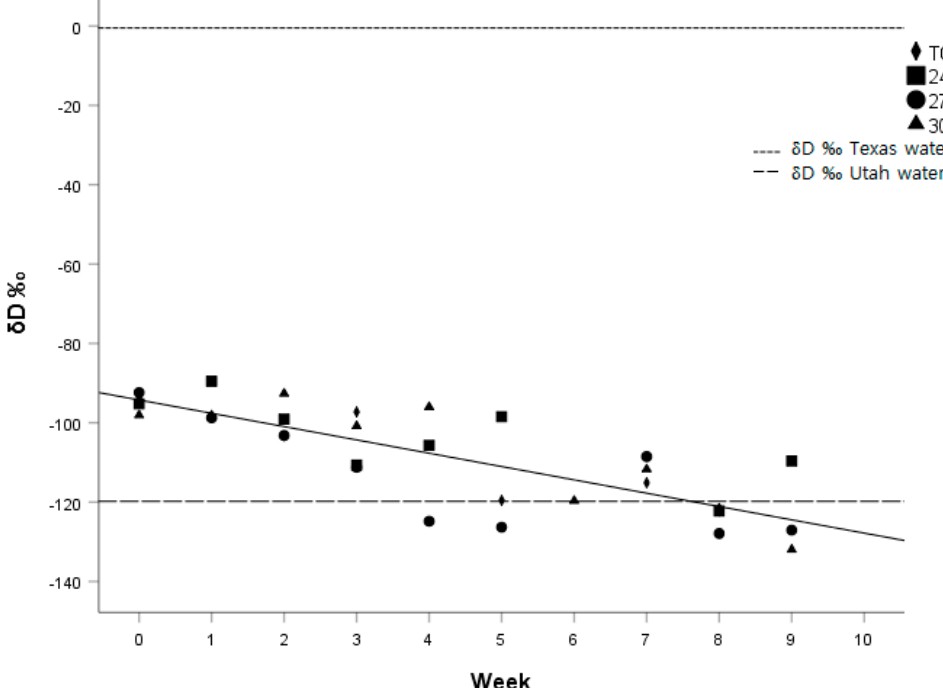

**Figure 3.** The $\delta$D ‰ values of scales of corn snakes (*Elaphe guttata*) for the three treatment groups and the thermoregulating group. Squares represent snakes held at a constant 24 °C, circles represent snakes held at a constant 27 °C, triangles represent snakes held at a constant 30 °C, and diamonds represent snakes allowed to freely thermoregulate. The short-dashed line represents the $\delta$D value of the water they were given previous to the experiment in Texas (−0.5‰) and the long-dashed line represents the $\delta$D value of the water they were given during the experiment in Utah (−119.8‰).

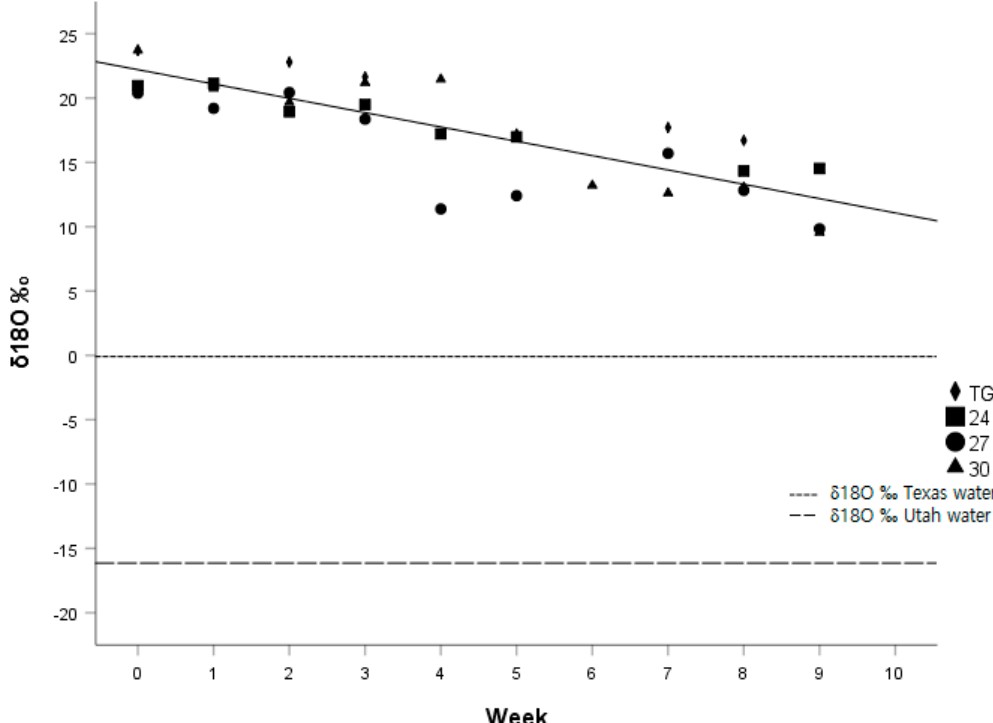

**Figure 4.** The $\delta^{18}O$ ‰ values of scales of corn snakes (*Elaphe guttata*) for the three treatment groups and the thermoregulating group. Squares represent snakes held at a constant 24 °C, circles represent snakes held at a constant 27 °C, triangles represent snakes held at a constant 30 °C, and diamonds represent snakes allowed to freely thermoregulate. The short-dashed line represents the $\delta D$ value of the water they were given previous to the experiment in Texas ($-0.85‰$) and the long-dashed line represents the $\delta D$ value of the water they were given during the experiment in Utah ($-16.15‰$).

The time span of our study (eight weeks) was not long enough to observe complete isotopic turnover in $\delta D$ and $\delta^{18}O$ in the scales of juvenile corn snakes. If we assume that the difference in the week 0 snakes and the $\delta D$ and $\delta^{18}O$ values of the Texas water is the fractionation value, and that the turnover of $\delta D$ and $\delta^{18}O$ values continue at the same linear rate as the regression equations of our calculated model beyond the eight weeks of our study, then we would expect the complete turnover of $\delta D$ and $\delta^{18}O$ in the scales of the snakes to be 28 weeks and 13 weeks from week 0, respectively. Although extrapolating our linear regression models should be interpreted with great caution, it appears that the $\delta^{18}O$ values in the scales of snakes turn over at a greater rate than the $\delta D$ values. However, this evidence is far from conclusive and warrants a lengthier study with a larger sample size.

Different tissues were shown to turn over at different rates in juvenile corn snakes [25]. Future analyses of other tissues (blood, plasma, muscle, liver, etc.), especially those with a major water component such as blood plasma, may reveal a more complete turnover within this time span (eight weeks). However, the rate of turnover of each of the tissues could possibly have been altered by offering more or less food and water to all the snakes. Budgetary and time constraints limited the tissues that we analyzed in this study, but future studies that incorporated analyses of multiple tissues could reveal different results based on the effect of temperature and metabolism on specific tissues.

Over the time and temperature ranges we considered, differences in metabolic rates due to temperature were not great enough to produce significant differences in the rate of turnover or incorporation of $\delta D$ and $\delta^{18}O$. This finding contradicted our hypothesis that increasing the temperature would increase the rate of turnover in tissues because of the correlation between metabolism and temperature as found in experimental models [29,30]. However, other tissues not analyzed in this study (blood, plasma, muscle, liver, etc.) may reveal more distinct rates of turnover among the treatment

groups. Additionally, a difference in rate of turnover could have been discovered if we used a larger sample size, or a greater range of temperatures.

We could have increased the growth and metabolism by feeding the snakes more than once a week, but we wanted to control for growth among each of our treatment groups. Therefore, the only limiting factor that differed between the treatment groups was temperature, and we could attribute differences in turnover among treatment groups to the temperature difference. As such, the lack of a difference between treatment groups may have been caused by keeping the amount of food fed to each snake constant across temperature treatments. Future experiments could hold snakes at constant temperature, but give them varying amounts of food to see if this is the case.

We controlled the isotopic signature of both the drinking water and food of these snakes by feeding them mice that drank the same water as the snakes. This accounts for all sources of hydrogen ingested by snakes; however, it does not account for all sources of oxygen. Snakes and other eukaryotic organisms use molecular oxygen during cellular respiration. Molecular oxygen is converted to water as it accepts electrons at the end of the electron transport chain and becomes part of the organism's total body water. In addition, oxygen is also released as a byproduct of cellular respiration in the form of carbon dioxide. Since the $\delta^{18}O$ of atmospheric oxygen should be the same in both Texas and Utah, we expected the effects of turnover to be dampened by the inspired atmospheric oxygen; thus, turnover of $\delta^{18}O$ among temperature groups would be different than that of $\delta D$. None of the temperature groups of $\delta D$ or $\delta^{18}O$ were significantly different from each other, perhaps because our sample sizes and range of temperatures were too small to detect a difference.

Our results showed a distinction between turnover in oxygen and hydrogen isotopes consistent with oxygen being more tied to metabolism than hydrogen. Size is directly related to the relative amount of oxygen derived from food/water or atmospheric oxygen. Larger animals with lower mass-specific metabolic rates will have a larger proportion of their oxygen intake from drinking water and food; by comparison, smaller animals will have a lesser proportion of their oxygen intake from drinking water [31]. Therefore, $\delta D$ is less likely to fluctuate with metabolic rate than $\delta^{18}O$ because $\delta D$ comes completely from the ingestion of food and water, and $\delta^{18}O$ comes from the ingestion of food, water, and atmospheric oxygen [22]. As expected, the $r^2$ values of our $\delta^{18}O$ treatment groups were larger than the $r^2$ values of our $\delta D$ treatment groups at $r^2 = 0.684$ and $r^2 = 0.568$, respectively, indicating less variance among $\delta^{18}O$ treatment groups than $\delta D$ treatment groups. Similarly, the slope of $\delta^{18}O$ turned over at a significantly faster rate than $\delta D$ when we normalized the data (see Figure 2). Snakes behaviorally modify their temperature according to diel activity, which in turn causes their metabolic rate to fluctuate. Our treatment groups were held at a constant temperature 24 hours a day, which meant they could not modify their body temperature or metabolic rate behaviorally.

The difference in turnover of hydrogen and oxygen isotopes in this experiment is analogous to results of experiments that use the doubly labeled water technique to measure field metabolic rates. In the doubly labeled water technique, oxygen and hydrogen atoms of water are labeled (with radio or stable isotopes) and then injected into an organism's bloodstream and allowed to equilibrate. After a period of time, a blood sample is taken measuring the change in the ratio of isotopes in the organism of both oxygen and hydrogen. The difference in the change in the oxygen isotope ratio is greater (because some of it is released as $CO_2$, and some of it is expelled as water loss) than the change in the hydrogen isotope ratio (because hydrogen is only being lost through water loss). The difference between the change in oxygen isotopes and the change in hydrogen isotopes is used to measure oxygen consumption and infer metabolic rate. One distinction between doubly labeled water experiments and this experiment is that doubly labeled water experiments measure the loss of an isotopic label from a novel water source over time, and our experiment was able to detect the incorporation of a novel isotopic label from both a water and food source over time. Our results are consistent with results of doubly labeled water experiments which show oxygen to turn over at a greater rate than hydrogen (see Figure 2).

The results of this experiment could possibly be replicated and modified to measure metabolic rates in the field under conditions where an organism is shifting from one source of food and water with a distinct isotopic signature to another, such as during long-distance migrations over deserts and oceans. Other than in doubly labeled water experiments, $\delta^{18}O$ and $\delta D$ were not previously used to determine field metabolic rates. Identifying an appropriate organism would have the following requirements: (1) it must be a species that feeds on isotopically distinct food sources, and (2) switches from one distinct food and water source to another in a (3) relatively short period of time (weeks). Ideally, a tissue with continuous growth such as hair or claws could be used to show a record of the turnover of isotopes over this time. The difference in the turnover of hydrogen and oxygen isotopes would also need to be determined under laboratory conditions (similar to this experiment). This method could provide valuable, long-term data on metabolic rate over species that temporally and/or spatially switch their diets, such as long-distance migratory species. To date, a study like this is yet to be done; however, potential migrating birds or bats could be identified using known migration routes and $\delta D$ or $\delta^{18}O$ databases [32,33].

The use of scales to determine turnover rates could be used as a non-invasive or minimally invasive method to collect isotopic data from organisms that regularly leave skin sheds, such as reptiles, insects, and amphibians. Skin sheds of snakes and other large ectotherms can often be found and collected for analyses of stable isotopes, or scales could be taken from specimens caught in the field. Analysis of the isotopes within skin sheds could reveal migration events, water source, and/or metabolic rate depending on the isotopes being analyzed.

More conclusive studies that build on our results are needed. Our study was not long enough to determine the time needed for growing juvenile corn snakes to equilibrate to a novel signature of $\delta D$ or $\delta^{18}O$ in their water and food. In addition, our temperature groups did not exhibit significant differences in the turnover rate of $\delta D$ or $\delta^{18}O$ in scales, again probably due to the short duration of the study. Studies that have larger sample sizes and measure other variables such as metabolic rate would be useful in furthering our understanding of $\delta D$ or $\delta^{18}O$ turnover in ectotherms.

**Author Contributions:** Conceptualization, K.A.H.; methodology, K.A.H.; software, S.J.H.; validation, S.J.H. and K.A.H.; formal analysis, S.J.H. and K.A.H.; investigation, S.J.H. and K.A.H.; resources, K.A.H.; data curation, S.J.H.; writing—original draft preparation, S.J.H.; writing—review and editing, S.J.H. and K.A.H.; visualization, S.J.H.; supervision, K.A.H.; project administration, S.J.H. and K.A.H.; funding acquisition, S.J.H.

**Funding:** Brigham Young University Office of Research and Creative Activities: Mentoring Grant.

**Acknowledgments:** We would like to thank the ORCA mentoring grant for funding this project, the University of Utah Stable Isotope Laboratory for analyzing the samples, and Brigham Young University for the use of its equipment and facilities. I would like to personally thank my family, especially my wife, Kimberley Hirt, for her unwavering love and support, and my parents, Mike and Martha Hirt. I would also like to thank all those who offered academic advice and support, including my graduate mentor Dr. Troy Best, and Bailey Brannan for her help in formatting the manuscript.

**Conflicts of Interest:** Kent A. Hatch is an author of this paper, as well as an editor for the journal of submission (Diversity).

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
