# Peer review of "The Effects of Temperature on the Turnover of δ18O and δD in Juvenile Corn Snakes (Elaphe guttata): A Novel Study with Ecological Implications"

_diversity, doi:10.3390/d11020019_

Round 1

Reviewer 1 Report

Typo on line 106 - Brigham Young University?

Given the temperature treatments used and the supposition that they impact metabolism/growth were any growth measurements determined?  For example, were snout-vent length or body mass determined following euthanasia? Examining growth rate interaction with the turnover rates would make sense given the treatments used.

Were air temperature treatments potentially confounded by the light source used?  This was not discussed but long and short wave radiation balance can significantly impact organism body temperature dominating the energy balance irrespective of a narrow air temperature. A description of refugia use by the snakes would be a nice addition to understanding the potential treatment effects, if available.

Baseline data are important to the issue of isotopic labeling in ectotherms given an isothermal environment. However, any comments that could be added to the manuscript describing the study environment would increase the applicability of the data to field measurements.

Author Response

Point 1: Given the temperature treatments used and the supposition that they impact metabolism/growth were any growth measurements determined?  For example, were snout-vent length or body mass determined following euthanasia? Examining growth rate interaction with the turnover rates would make sense given the treatments used.

Response 1: We did take mass measurements at the time of euthanasia, however we neglected to make measurements throughout the study so a change in mass was not able to be measured. This neglected oversight would have been valuable data, and I will add the following to the methods and discussion.

Methods: Mass at death was also measured before euthanasia, but we neglected to make mass measurements throughout the study so an analysis on the change in mass was not possible.

Discussion: Variables for change in growth were not included in our analyses, which would be an important factor to include in future studies.     

Point 2: Were air temperature treatments potentially confounded by the light source used?  This was not discussed but long and short wave radiation balance can significantly impact organism body temperature dominating the energy balance irrespective of a narrow air temperature. A description of refugia use by the snakes would be a nice addition to understanding the potential treatment effects, if available.

Response 2: I added the following to the methods description in the descriptions of the habitats including the light source and the placement of the lights in relation to the snakes and their refugia.

Each snake in a temperature controlled group was housed in an 8x12x4 inch sized plastic container with holes in the lids for ventilation and wood chips within the container to provide some structural habitat and refugia. The same type of lighting (fluorescent bulbs) were used within the temperature cabinets and in the freely thermoregulating group, and differences between light exposures were minimized by randomizing the location of each snake during watering and feeding events. Lights were put on a 12 hour light and dark alternating timer. The lights in the temperature cabinets were positioned along the sides of the cabinet extending vertically, while the lights of the thermoregulating group were placed on top of the cages extending horizontally. 

Point 3: Baseline data are important to the issue of isotopic labeling in ectotherms given an isothermal environment. However, any comments that could be added to the manuscript describing the study environment would increase the applicability of the data to field measurements.

Response 3: I believe the changes to responses 2 made in the manuscript also account for the suggestion to this point as well. 

I added the following to the methods description in the descriptions of the habitats including the light source and the placement of the lights in relation to the snakes and their refugia.

Each snake in a temperature controlled group was housed in an 8x12x4 inch sized plastic container with holes in the lids for ventilation and wood chips within the container to provide some structural habitat and refugia. The same type of lighting (fluorescent bulbs) were used within the temperature cabinets and in the freely thermoregulating group, and differences between light exposures were minimized by randomizing the location of each snake during watering and feeding events. Lights were put on a 12 hour light and dark alternating timer. The lights in the temperature cabinets were positioned along the sides of the cabinet extending vertically, while the lights of the thermoregulating group were placed on top of the cages extending horizontally. 

In addition to the changes made above, more detail was provided about the allocation of drinking water (via petri dishes) in the methods section. 

Reviewer 2 Report

This manuscript describes a study to investigate the kinetics of 2H and 18O changes following a water and diet switch between at hatching over 8 weeks.

The snakes were subjected to one of three constant temperatures or a thermal gradient over the study. Researchers found that 18O signals in the scales changed faster than the 2H and that the constant temperatures caused a faster change in 18O than the snakes given the thermal gradient.

This is an interesting study and the first of its kind with respect to examining H and O turnover following ‘diet-switches’. I have no major issues with the ms but suggest (below) several things that can make the ms more impactful and clearer to the reader. Other comments involve corrections or clarifications of fact.

Line 45: I disagree that ectotherms no not need to maintain elevated Tb; particularly in the case of improving locomotion speed  (for prey capture or predator escape, mating, reproduction rates, improving digestion rates, etc.). Please clarify.

Line 46: Having a ‘lower’ metabolic rate is not necessarily a more ‘efficient’ metabolic rate. Please clarify.

Line 53: This statement is NOT true when animals are within or above their TNZ.

Line 74: This is a strange/confusing sentence structure.

Line 79: The fact that this species breeds year-round does not seem an important criterion.

Line 93-96: Please specify the 2H and 18O values here – yes, I see that they are reported later in the Results section.

Line 106: Check spelling of Young

Line 114: Why didn’t the authors do ‘non-lethal’ sampling and sampling of shed tissues/skins since this is the way they propose future researchers use this technique. Furthermore, it was unfortunate that the authors did not examine the many other body tissues to make their proposed mechanisms more robust.

Line 204: What is ‘coefficient of time’; this term is unclear.

Figures (In general): It would be very helpful if the authors labeled the functions for the readers. The information is in the caption but would be far more impactful if they were labeled directly (there is certainly sufficient white-space to do this).

Figure 4: There is no need for decimal values to the hundredths. These should be eliminated.

MS (in general): Check formatting of “CO2

Author Response

Point 1:Line 45: I disagree that ectotherms no not need to maintain elevated Tb; particularly in the case of improving locomotion speed  (for prey capture or predator escape, mating, reproduction rates, improving digestion rates, etc.). Please clarify.

Response 1: I changed the line to reflect the point that ectotherms do not need to have a constantly elevated temperature and can vary according to activity as noted by the reviewer

Additionally, ectotherms have no metabolic need to maintain a constant and continual elevated body temperature, allowing them to generally have a more efficient metabolism then endotherms wherein they can behaviorally match their metabolic need with environmental temperatures. 

Point 2 Line 46: Having a ‘lower’ metabolic rate is not necessarily a more ‘efficient’ metabolic rate. Please clarify.

Response 2: Being able to match the metabolic need to environmental temperatures is more generally efficient than using metabolism to regulate body temperature within a narrow range of temperatures. I changed the wording of the sentence to reflect that difference.

Additionally, ectotherms have no metabolic need to maintain a constant and continual elevated body temperature, allowing them to generally have a more efficient metabolism then endotherms wherein they can behaviorally match their metabolic need with environmental temperatures. 

Point 3 Line 53: This statement is NOT true when animals are within or above their TNZ.

Response 3: I added the qualifier "below their thermoneutral zone" for clarification

Point 4 Line 74: This is a strange/confusing sentence structure.

Response 4: I restructured the sentence. 

We tested the hypothesis that δD and δ18O in scales of snakes will turnover at a greater rate when snakes are held at higher temperatures. 

Point 5 Line 79: The fact that this species breeds year-round does not seem an important criterion.

Response 5: "breeds year-round" was changed to "readily available"

Point 6:Line 93-96: Please specify the 2H and 18O values here – yes, I see that they are reported later in the Results section.

Response 6: The values were inserted as suggested

Point 7 Line 106: Check spelling of Young

Response 7: Spelling corrected

Point 8 Line 114: Why didn’t the authors do ‘non-lethal’ sampling and sampling of shed tissues/skins since this is the way they propose future researchers use this technique. Furthermore, it was unfortunate that the authors did not examine the many other body tissues to make their proposed mechanisms more robust.

Response 8: We had hoped to analyze more tissues to give a more complete analysis on the turnover, but we lacked the funding to do so. The following line was added to the manuscript in the discussion

"Budgetary and time constraints limited the tissues that we analyzed in this study, but we hope to analyze more tissues for comparison in future studies."

Point 9 Line 204: What is ‘coefficient of time’; this term is unclear.

Response 9: "the coeffecient of" deleted so the sentence now reads "As expected, time had a significant effect on the turnover of δD and δ18O in the scales of our juvenile corn snakes"

Point 10 Figures (In general): It would be very helpful if the authors labeled the functions for the readers. The information is in the caption but would be far more impactful if they were labeled directly (there is certainly sufficient white-space to do this).

 Response 10: Symbols and lines in the figures have been labelled to make the information more obvious and intuitive

Point 11 Figure 4: There is no need for decimal values to the hundredths. These should be eliminated.

 Response 11: The decimals have been removed from the figure

Point 12 MS (in general): Check formatting of “CO2”

Response 12: I have found typos associated with the formatting of CO2 and rewritten with the 2 as a subscript.

Reviewer 3 Report

A well presented paper that uses a novel approach to characterize corn snake origins and growth cycles using isotope analyses.

The scientific analyses appear sound and the statistical quantification appropriate

Only minor edits are required;

Page 1 line 11 insert of after effects

Page 1 abstract line 12: You use upper case C for Corn snakes when it should be lowercase.

Page 3 line 94 it should ensure not insure

Author Response

Point 1: Page 1 line 11 insert of after effects

Response 1: "of" inserted after the word "effects"

Point 2: Page 1 abstract line 12: You use upper case C for Corn snakes when it should be lowercase.

Response 2: typo corrected

Point 3: Page 3 line 94 it should ensure not insure

Response 3: correction made